# Strengthening Tuberculosis Control Among Migrant Workers

**DOI:** 10.3390/tropicalmed9110274

**Published:** 2024-11-12

**Authors:** Saurabh RamBihariLal Shrivastava, Prateek Sudhakar Bobhate, Prithvi Brahmanand Petkar, Harshal Gajanan Mendhe, Gulshan Ruprao Bandre

**Affiliations:** 1Department of Community Medicine, Datta Meghe Medical College, Off-Campus Centre of Datta Meghe Institute of Higher Education and Research, Nagpur 441110, Maharashtra, India; prithvip76@gmail.com (P.B.P.); drharshalmendhe@gmail.com (H.G.M.); 2Department of Community Medicine, All India Institute of Medical Sciences, Vijaypur 180001, Jammu, India; prateekbobhate@gmail.com; 3Department of Microbiology, Jawaharlal Nehru Medical College, Datta Meghe Institute of Higher Education and Research, Sawangi (M), Wardha 442005, Maharashtra, India; gulshanbandre21@gmail.com

**Keywords:** tuberculosis, migrant, health, screening, stigma

## Abstract

Tuberculosis (TB) is a serious infectious disease accounting for a significant number of deaths due to the infectious nature of the disease on the global platform. Migrant workers need special attention as these population groups live in substandard and crowded environmental conditions with poor ventilation, which play a crucial role in augmenting the risk of acquisition of infection. The global vision to ensure the delivery of effective TB control-related services for migrant workers has been influenced by a wide range of barriers. This issue is further complicated by the limited knowledge of migrant workers about tuberculosis, their rights, the kind of services available in healthcare facilities, and the ways to prevent the acquisition and transmission of infectious disease. By acknowledging the role of predisposing factors and the potential barriers that impact accessing timely healthcare services, it can be seen that the need of the hour is to plan and implement a comprehensive package of services for the benefit of migrant workers.

## 1. Introduction

Tuberculosis (TB) is a serious infectious disease accounting for a significant number of deaths due to the infectious nature of the disease on the global platform [1]. It is an alarming fact that, despite being a preventable and treatable disease, a total of 1.3 million people have lost their lives due to tuberculosis-associated complications [2]. In addition, a total of 10.6 million people were diagnosed with the disease in 2022, with the majority of cases and deaths being reported from low- and middle-income nations where there are multiple other competing health priorities and healthcare systems that are not well-equipped to effectively contain and manage the disease [1,2]. Moreover, the multidrug- and severely drug-resistant forms of TB have the potential to overwhelm any health sector due to their complexity and rising treatment costs [3]. Furthermore, the disease has resulted in major financial losses, reduced productivity, and increased direct and indirect healthcare expenditures [4].

## 2. Tuberculosis and Migrant Workers

Even though the disease affects people from all age groups, people from specific groups are at higher risk of acquiring the infection, such as migrant workers, prisoners, people with a coinfection of human immunodeficiency virus, people with other comorbidities, etc. [5,6]. The estimated incidence of TB among the migrant population in 30 low-incidence nations varied between 24 million in Iceland to 100 million in the United Kingdom [7]. Migrant workers need special attention as these population groups live in substandard and crowded environmental conditions with poor ventilation, which play a crucial role in augmenting the risk of acquiring an infection [8,9]. There is a definite need to acknowledge that there are different types of migrant workers, and this has been mentioned in Table 1 [5,6,8].

Significant differences also exist depending on the sector in which they work, whether formal or informal, such as in the mining, domestic, or agricultural industries, as well as regional- and country-specific factors that must be acknowledged [4,5,6,8]. Moreover, the infection directly impacts the earning ability of these workers, and the treatment cost can push them below the poverty line (catastrophic expenditure) [10,11]. Further, as these people continue to move from one place to another, it becomes extremely difficult for healthcare providers to deliver treatment and ensure a complete follow-up [12]. In addition, more often than not, these workers are often exposed to occupational hazards at their workplace making them extremely susceptible to acquiring the infection [13]. At the same time, these workers might have varied cultural barriers that can also influence treatment adherence practices [14,15].

## 3. Predisposing Factors

The incidence and prevalence of tuberculosis among migrant workers is extremely high and this is because of the simultaneous presence of multiple factors [8]. The presence of low socioeconomic statuses among migrant employees tends to influence their nutritional status and housing conditions, both of which account for heightened susceptibility [16]. As mentioned earlier, poor housing standards, including overcrowding and poorly ventilated homes, are extremely important in influencing the transmission of this airborne infection [8,10]. Owing to the absence of documentary proof regarding their origin and address, these workers have limited access to healthcare services, which accounts for delays in diagnosis and the initiation of appropriate treatment [12]. As migrant workers often do not have fixed addresses in the government records, they often do not seek medical care due to the fear of being sent back to their original place or being subjected to legal concerns [17]. Vitamin D deficiency has been identified as one of the risk factors for TB infection among people migrating from Africa and Asia to the Western nations [18,19].

At this juncture, we cannot rule out the possible impact of having a fear of stigma and discrimination, which discourages them from seeking timely medical help (as it is bound to impact their vocational opportunities) [20,21]. There is a definite possibility that these workers will have limited knowledge about the symptoms and mode of transmission of the infection, the significance of treatment adherence, and the necessity to complete the treatment [21]. This is bound to impact every aspect of the management of this highly infectious disease. One of the major factors that results in treatment failure or the development of resistance among these workers is their habit of frequently moving between different geographical locations, which accounts for interruptions in the course of treatment [8]. Further, there is a major barrier of cultural insensitivity or language barriers that influence the quality of communication between the workers and healthcare professionals; thus, migrant workers often fail to comprehend the provided information [14,15].

On the occupational front, there is proven evidence suggesting that people who are exposed to silica dust tend to have a higher risk of acquiring TB infection, and this is quite common among migrant employees [13]. These workers tend to have inadequate nutrition which in turn can impair their immune response, thereby augmenting the possibility of acquiring a TB infection and the progression of the disease [8,22]. We must not ignore the presence of coinfections and comorbidities, as the simultaneous presence of both these conditions can exacerbate the vulnerability to acquiring the infection [23]. Furthermore, in most nations, there are no policies or programs targeting the health standards of migrant workers, which can play a defining role in complicating the management of TB among migrant workers [24,25,26]. The presence of all the above predisposing factors in varied combinations can significantly increase the susceptibility of migrant workers to acquiring the infection or expedite the progression of the disease [10,21,24,26,27].

## 4. Barriers to Effective TB Control Among Migrant Workers

The global vision to ensure the delivery of effective TB control-related services for migrant workers has been influenced by a wide range of barriers. In order to keep the entire system of TB control accountable, the address of the patient and a supporting government document are often looked upon as the initial parameter on which the decision of healthcare providers to initiate and continue treatment is based [12,17]. In the case of migrant workers, as they keep moving from one place to another, more often than not, they do not have a document supporting their address, and this becomes a major barrier to treatment initiation and completion [12,17]. In addition, from a legal perspective, there is always a fear of deportation, especially when migrants have traveled to adjoining nations, and thus they feel there is an immense risk in seeking healthcare services [25,26]. Furthermore, the possibility of discrimination from health workers can also not be ruled out, and this further discourages migrant workers from seeking healthcare services despite being symptomatic [28].

The next major barrier is the high cost that has been linked with the treatment, which is often of long duration [4,10]. As government officials often ask for proof of address, the migrant workers approach private healthcare providers, and soon, once they start feeling better (after initial treatment), they discontinue treatment because of their inability to afford treatment [4,8,10]. This is a major hurdle for migrant workers as money spent on the diagnosis and treatment can easily push them below the poverty line. In contrast to other people, migrant workers do not have health insurance coverage and thus it is just not possible for them to bear the out-of-pocket expenditures associated with treatment for the entire course of therapy [4,10]. There is another risk that if these workers are diagnosed with the infection, there is a definite possibility that these workers will be subjected to stigma and discrimination, including losing their jobs, which can prove to be extremely dangerous for their families [28].

Migrant workers tend to have different cultural beliefs and expectations which may not be in alignment with the medical practices in the host country, and this can significantly influence the practice of treatment adherence [14,21]. In addition, the presence of a language barrier can greatly impact the quality of communication between healthcare providers and migrant workers [29]. In fact, the absence of effective communication plays a vital role in the lack of treatment adherence and discontinuation of therapy mid-way through [29]. On the job front, these workers often have long working hours and no provision to avail of paid sick leave, and thus these workers continue to delay seeking healthcare services [12,27]. At the same time, due to the dependence on employers for any kind of healthcare information and lack of support for accessing healthcare, these workers are deprived of timely healthcare and this plays a crucial role in the advancement of the disease [21]. In addition, the rigid timings of the public health sector and unrealistic working timings of migrant workers often negatively impact their desire to access healthcare [30].

This issue is further complicated by the limited knowledge that migrant workers have about tuberculosis, their rights, the kind of services available in healthcare facilities, and the ways to prevent the acquisition and transmission of infectious disease [14,28]. Barriers have been identified even from the healthcare delivery system side, wherein, owing to the complexities in the working mechanism and fear of authorities, many migrant workers are reluctant to access healthcare [31,32]. Further, some healthcare facilities lack the basic infrastructure to effectively manage TB patients, and this makes it quite challenging for the migrant workers to avail of healthcare services [31,32]. In fact, a lack of reliable transportation options can significantly hinder access to healthcare services. Furthermore, we also cannot rule out the possibility of prior bad experiences owing to which these vulnerable population groups have limited trust in healthcare providers [12].

In addition, the absence of health-related information in their native language can also prove to be a significant barrier as they cannot comprehend the available information, education, and communication materials [29]. The reality is that there is limited advocacy for the health needs of migrant population groups, and thus their needs are often ignored and not given importance on any platform, including while framing policies or regulations [24,25,26]. It would not be an understatement that migrant population groups lack access to any social networks or welfare agencies, and thus neither themselves nor their children receive any kind of support if they acquire tuberculosis infection [33]. This could be due to the stigma related to TB, which can result in social isolation, reluctance to seek healthcare, and subsequent delays in diagnosis and treatment [33]. Moreover, the issues of frequent movement, substandard living conditions, and mental stigma have also emerged as major barriers not permitting migrant workers to timely avail of healthcare services [20,34]. These constitute the wide spectrum of challenges and barriers that influence and impact migrant workers to take care of their health and well-being [12,21].

## 5. Strategies for Strengthening TB Control

By acknowledging the role of predisposing factors and the potential barriers that impact accessing timely healthcare services, it can be seen that the need of the hour is to plan and implement a comprehensive package of services for the benefit of migrant workers. Depending on the characteristics that are specific to different categories of migrant workers, internal, international, irregular, and low-wage groups of migrant workers are the most vulnerable to TB and therefore warrant special attention. With reference to international, irregular, and low-wages categories of migrant workers, there is an immense need to strengthen health education and awareness activities by resorting to tailor-made education campaigns [35]. The purpose of such a campaign is to create awareness about the disease and this can be executed with the help of developing culturally sensitive educational material, increasing the availability of learning resources in different languages, and distributing educational materials through different platforms (viz. social media) and in different settings (like community or workplace) [29,36]. These materials should predominantly target the symptoms of disease, modes of transmission, prevention strategies, and the necessity to complete the entire course of treatment [29,36]. It is always encouraged to include images and visual aids to maximize our reach to illiterate population groups, and this strategy could prove vital for the low-wage group of migrant workers.

For the group of international, irregular, and low-wage migrant workers, it is advisable to involve members of the community and peers as educators, and this can be accomplished by training them for their role and using them in community events [37]. Even non-governmental agencies can be involved in these awareness activities, especially in those areas that lack government or private healthcare services, and this will definitely help all groups of migrant workers who do not have easy access to healthcare providers. The next important area that needs attention for migrant workers is the strengthening of screening and early detection activities [38]. This can be accomplished by the establishment of mobile clinics and outreach programs to reach remote or underserved locations, which generally harbor internal, international, irregular, and low-wage categories of migrant workers, with or without the assistance of local organizations [39,40]. However, these mobile units must visit those places where migrant workers live or work on a regular basis to ensure continuity of care. At the same time, it is important that TB screening should be made a part of general health check-ups, preferably using rapid diagnostic tests for quicker results, and this strategy should be implemented for all types of migrant workers [38]. In-fact, screening for tuberculosis has been initiated as a regular practice across different nations among migrant populations [41,42,43]. Furthermore, testing for the disease should be made mandatory regardless of migrant worker categories, as encouraging participation it is vital to ensuring confidentiality [44].

The diagnostic services should be followed by strengthening the treatment and care offered to diagnosed patients. From the treatment perspective, barring high-skilled migrant workers who have better access to healthcare services and health insurance, the healthcare delivery system must aim to ensure uninterrupted treatment and follow-up and this will essentially depend on the implementation of directly observed therapy, giving flexible treatment options to workers to accommodate their work timings, and exploring the possibility of providing support for transport or some kind of financial aid to the diagnosed patients [34,45,46]. In addition, a mechanism must be built to ensure a seamless transition of care for all categories of migrant workers who do not have proper documentation, as migrant patients may decide to move to another place making it difficult for them to obtain treatment in the absence of a proof of address document [47]. Moreover, apart from treatment, for international migrant workers, we must aim to provide culturally and linguistically appropriate and sensitive care to the diagnosed migrant cases, and this starts with training healthcare providers in cultural competencies and helping both parties with translation services [14,15]. Healthcare providers must respect the cultural practices and beliefs of these vulnerable population groups and at the same time develop education materials in different languages to meet their needs [14,15].

The next important area is strengthening laws to safeguard the health rights of international and irregular migrant workers and this requires implementing policies to promote access to healthcare for all migrants, safeguarding them against all kinds of discrimination in healthcare settings, and providing legal support to them, if required [24,48]. Even in the workplace settings, a specific set of workplace policies should be implemented to facilitate periodic health check-ups and treatment, and this is a must for internal, international, regular, irregular, and low-wages group of migrant employees [25]. As migrants can cross even international boundaries, which is the case with international migrant workers, there has to be collaboration between nations and regions to ensure that coordinated care and services are offered to patients, including the sharing of data and best practices [25,26]. Finally, the prevention and control services can be further augmented by promoting research activities in the domain of the prevalence and control of the disease among different migrant population groups [49,50,51]. These studies can target epidemiological aspects and social determinants influencing TB risk among migrants to evaluate the impact of different interventions and explore the potential barriers to accessing TB care, etc. The World Health Organization has created a set of guidelines for strengthening the social protection of these vulnerable population groups by making the existing programs more responsive to the needs of the patients of concern, promoting the engagement of relevant ministries and stakeholders, to address income and food safety needs, including provision of nutritional support [52]. Furthermore, the data obtained from such research work should be used to frame evidence-driven policies and interventions [49,50,51].

## 6. Case Studies

A number of innovative interventions have been planned and implemented in different parts of the world to strengthen TB prevention and control services. In the Kabul city of Afghanistan, to which a large number of migrants have migrated in search of better opportunities and due to internal conflicts, the Médecins Sans Frontières (MSF) has initiated the services of preventive mobile clinics to provide access to healthcare services, including screening for tuberculosis to promote early detection and the timely initiation of treatment to avoid development of complications [53]. Such kinds of services offered by MSF can be of immense help for internal, irregular, and low-wage groups of migrant employees. Active case finding is an effective strategy to promote early detection among migrant population groups [54]. The employment of mobile health (mHealth) applications has been encouraged among migrant population to strengthen different aspects of tuberculosis treatment, such as individualizing dosages, monitoring adherence to treatment regimen, and creating awareness about the disease [55]. These mHealth applications have enormous potential to streamline treatment for internal, international, irregular, low-wage, and even regular groups of migrant workers. Adherence monitoring has been carried out either through synchronous or asynchronous video directly observed treatment, using applications like miDOT-EMOCHA, SureAdhere, and AiCure [55]. These adherence monitoring applications can be used by different types of migrant workers.

With India being a high-burden nation that contributes almost a quarter of the overall global TB cases, it has been reiterated time and again that the public health sector alone cannot bridge the existing gap. In nations like India and Brazil, conditional cash transfer schemes have been introduced in different areas of TB treatment, and this has had a positive influence on improving the success rates of treatment outcomes [56,57]. Such conditional cash transfer schemes can be of great benefit to migrants with a poor socioeconomic status (viz. internal, international, irregular, and low-wage). In fact, to support the government and authorities, a number of non-governmental organizations have been founded to reach out to the migrant population, who generally encounter a wide range of barriers. For instance, the Aajeevika organization aims to provide basic support to migrants by creating proper documents for them, which are required to avail of healthcare services [50]. These organizations can definitely help irregular and international groups of migrant workers as they always have a fear of deportation. Similarly, another non-governmental organization, PRAYAS, has been working to support different groups of migrant workers to come together, be organized, and put forth an appeal to their employers to raise their salaries [50]. Organizations like PRAYAS can significantly improve the working conditions of internal, international, regular, irregular, and low-wage migrant workers who are more often than not exposed to occupational hazards. In addition, this organization is also proactive in India to promote the access of migrant workers to healthcare services. Similar kind of initiatives have also been undertaken in Myanmar to help migrant workers [50].

## 7. Conclusions

In conclusion, migrant workers from different categories are vulnerable to acquiring tuberculosis infection due to the presence of multiple coexisting factors both in the domestic and workplace settings. However, as these migrant populations encounter a wide range of barriers to accessing TB care-related services, the need of the hour is to adopt a comprehensive package of targeted interventions to improve the existing scenario and improve treatment outcomes among them.

## Figures and Tables

**Table 1 tropicalmed-09-00274-t001:** Migrant worker categories and their vulnerability to TB.

Migrant Worker Categories	Description	Vulnerability to TB
Internal	Workers who move within their own country, often from rural to urban areas, in search of better employment opportunities	Poor housing conditions Limited access to healthcare Occupational hazards Mobility Stigma and discrimination
International	Workers who cross national borders to find employment, often due to conflicts, or natural disasters	Legal and administrative barriers (no documents) Language and cultural barrier Fear of deportation Exposure to causative organisms during transit
Regular	Workers who migrate legally and possess the required documentation and permits to work in a foreign country	Access to healthcare Poor workplace conditions No health insurance Discrimination from local people
Irregular	Workers who migrate without legal authorization, and do not have official documents and rights in the host country	Lack of access to healthcare Poor workplace conditions Fear of deportation Risk of exploitation Lack of social support
High skilled	Workers who migrate for employment in highly skilled professions (viz. Doctors, Academicians, etc.)	Better income, often with insurance coverage Generally safe and better regulated working conditions High health literacy Low risk due to better living conditions and access to healthcare Exposure risk during travel Work-related stress might lower immunity
Low wage	Workers engaged in low-paying jobs (like agriculture, construction, household work, etc.)	Lower income, often without health insurance Physically demanding and poorly regulated working conditions Poor health literacy Poor housing conditions Occupational exposures Limited access to healthcare Poor nutritional status

## Data Availability

The literature pertaining to the article can be made available at request by contacting Saurabh Shrivastava at drshrishri2008@gmail.com on reasonable request.

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
