# Peer review of "Strengthening Tuberculosis Control Among Migrant Workers"

_tropicalmed, 2024, doi:10.3390/tropicalmed9110274_

Round 1

Reviewer 1 Report

Comments and Suggestions for Authors

The topic is of critical importance. Several suggestions are presented for the authors to consider:

  1. Table 1 outlines the definition of migrant workers. However, the categories "high-skilled" and "low-wage" may intersect with other categories. The authors should consider incorporating additional explanations into the text to clarify these overlaps.
  2. In light of the previously mentioned point, it would be beneficial to discuss the 'predisposing factors' and 'barriers to effective tuberculosis (TB) control' specifically in relation to different categories of migrant workers. Furthermore, the sentence found on lines 158-160 seems to lack coherence with the surrounding context. Could there be an emphasis placed on the challenges faced by the migrant workers or their family members who are in close contact?
  3. The outlined "strategies for strengthening TB control" come across as overly general. As previously indicated, a more targeted approach would be valuable, particularly one that details how to mobilize resources for different categories of migrant workers.
  4. Additionally, the case study could be utilized to showcase exemplary methods of targeting various categories of migrant workers. Thus, the conclusion could be improved through those specific analyses.

Author Response

Reviewer 1

The topic is of critical importance. Several suggestions are presented for the authors to consider:

 Comment 1: Table 1 outlines the definition of migrant workers. However, the categories "high-skilled" and "low-wage" may intersect with other categories. The authors should consider incorporating additional explanations into the text to clarify these overlaps.

Response 1: Thank you for pointing this out. We agree with the comment and accordingly we have added additional points of differentiation in Table 1 (Page 2 and Page 3 – Red colour)

Comment 2: In light of the previously mentioned point, it would be beneficial to discuss the 'predisposing factors' and 'barriers to effective tuberculosis (TB) control' specifically in relation to different categories of migrant workers. Furthermore, the sentence found on lines 158-160 seems to lack coherence with the surrounding context. Could there be an emphasis placed on the challenges faced by the migrant workers or their family members who are in close contact?

Response 2: Thank you for pointing this out. We have re-written the sentence and made it logical for the readers to understand it (Page 4, line 162-166)

Comment 3: The outlined "strategies for strengthening TB control" come across as overly general. As previously indicated, a more targeted approach would be valuable, particularly one that details how to mobilize resources for different categories of migrant workers.

Response 3: Thank you for pointing this out. We agree with the comment and made the entire sub-heading with reference to the different group of migrant worker categories (Page 5, line 174-177; Line 183-185; Line 186-191; Line 193-197; Line 198-200; and Page 6, line 201-204, 206-214, Line 215-217; Line 222-223, Line 226-234)

Comment 4: Additionally, the case study could be utilized to showcase exemplary methods of targeting various categories of migrant workers. Thus, the conclusion could be improved through those specific analyses.

Response 4: Thank you for pointing this out. We agree with the comment and case studies and conclusion has been accordingly revised. (Page 6, line 250 to Page 7, line 253; Page 7, line 256-258; Line 260-261; Line 266-268, Line 273-274; Line 277-279, Line 283-385)

Reviewer 2 Report

Comments and Suggestions for Authors

The manuscript addresses an importance issue: Tuberculosis control among migrant worker population. With the aim to improve the study, some suggestions could be indicate:

1. A quantitative approach, considering the incidence and prevalence of tuberculosis in migrant populations from different countries may be useful to understand the public health problem (1). 

2. A risk factor of tuberculosis in migrant population from Africa and Asia countries to Western countries is vitamin D deficiency. This issue could be mentioned (2-3).

3. A general approach to migrants and tuberculosis is the study of Dhavan and co-authors (4) which may be appropriate.

4. Examples of screening programs of the tuberculosis control in migrant populations could be interesting (5-7).

5. A review for active case finding for tuberculosis in migrant populations could be mentioned (8).

References

1.Lönnroth K, Mor Z, Erkens C, Bruchfeld J, Nathavitharana RR, van der Werf MJ, Lange C. Tuberculosis in migrants in low-incidence countries: epidemiology and intervention entry points. Int J Tuberc Lung Dis. 2017;21:624-637. doi: 10.5588/ijtld.16.0845.

2.Lips P, de Jongh RT. Vitamin D deficiency in immigrants. Bone Rep. 2018;9:37-41. doi: 10.1016/j.bonr.2018.06.001.  

3.Hayward S, Harding RM, McShane H, Tanner R. Factors influencing the higher incidence of tuberculosis among migrants and ethnic minorities in the UK. F1000Res. 2018 13;7:461. doi: 10.12688/f1000research.14476.2.

4.Dhavan P, Dias HM, Creswell J, Weil D. An overview of tuberculosis and migration. Int J Tuberc Lung Dis. 2017;21:610-623. doi: 10.5588/ijtld.16.0917.

5.Silva DR, Mello FCQ, Johansen FDC, Centis R, D'Ambrosio L, Migliori GB. Migration and medical screening for tuberculosis. J Bras Pneumol. 2023;49:e20230051. doi: 10.36416/1806-3756/e20230051.

6 .D'Ambrosio L, Centis R, Dobler CC, Tiberi S, Matteelli A, Denholm J, Zenner D, Al-Abri S, Alyaquobi F, Arbex MA, et al. Screening for tuberculosis in migrants: A Survey by the Global Tuberculosis Network. Antibiotics (Basel). 2021;10:1355. doi: 10.3390/antibiotics10111355.

 7.Braga S, Vieira M, Barbosa P, Ramos JP, Duarte R. Tuberculosis screening in the European migrant population: a scoping review of current practices. Breathe (Sheff). 2024;20:230357. doi: 10.1183/20734735.0357-2023.

 8.Pramono JS, Ridwan A, Maria IL, Syam A, Russeng SS, Syamsuar, Mumang AA. Active case finding for tuberculosis in migrants: a systematic review. Med Arch. 2024;78(1):60-64. doi: 10.5455/medarh.2024.78.60-64.

 Author Response

Reviewer 2

The manuscript addresses an importance issue: Tuberculosis control among migrant worker population. With the aim to improve the study, some suggestions could be indicate:

Comment 1: A quantitative approach, considering the incidence and prevalence of tuberculosis in migrant populations from different countries may be useful to understand the public health problem (1). 

1.Lönnroth K, Mor Z, Erkens C, Bruchfeld J, Nathavitharana RR, van der Werf MJ, Lange C. Tuberculosis in migrants in low-incidence countries: epidemiology and intervention entry points. Int J Tuberc Lung Dis. 2017;21:624-637. doi: 10.5588/ijtld.16.0845.

Response 1: Thank you for pointing this out. We agree with the comment and it has been incorporated in the manuscript (Page 2, line 50-51)

Comment 2: A risk factor of tuberculosis in migrant population from Africa and Asia countries to Western countries is vitamin D deficiency. This issue could be mentioned (2-3).

2.Lips P, de Jongh RT. Vitamin D deficiency in immigrants. Bone Rep. 2018;9:37-41. doi: 10.1016/j.bonr.2018.06.001.  

3.Hayward S, Harding RM, McShane H, Tanner R. Factors influencing the higher incidence of tuberculosis among migrants and ethnic minorities in the UK. F1000Res. 2018 13;7:461. doi: 10.12688/f1000research.14476.2.

Response 2: Thank you for pointing this out. We agree with the comment and it has been added in the article (Page 3, line 81-82)

Comment 3: A general approach to migrants and tuberculosis is the study of Dhavan and co-authors (4) which may be appropriate.

4.Dhavan P, Dias HM, Creswell J, Weil D. An overview of tuberculosis and migration. Int J Tuberc Lung Dis. 2017;21:610-623. doi: 10.5588/ijtld.16.0917.

Response 3: Thank you for pointing this out. We agree with the comment, and it has been added in the article (Page 2, line 52-54)

Comment 4: Examples of screening programs of the tuberculosis control in migrant populations could be interesting (5-7).

5.Silva DR, Mello FCQ, Johansen FDC, Centis R, D'Ambrosio L, Migliori GB. Migration and medical screening for tuberculosis. J Bras Pneumol. 2023;49:e20230051. doi: 10.36416/1806-3756/e20230051.

6 .D'Ambrosio L, Centis R, Dobler CC, Tiberi S, Matteelli A, Denholm J, Zenner D, Al-Abri S, Alyaquobi F, Arbex MA, et al. Screening for tuberculosis in migrants: A Survey by the Global Tuberculosis Network. Antibiotics (Basel). 2021;10:1355. doi: 10.3390/antibiotics10111355.

7.Braga S, Vieira M, Barbosa P, Ramos JP, Duarte R. Tuberculosis screening in the European migrant population: a scoping review of current practices. Breathe (Sheff). 2024;20:230357. doi: 10.1183/20734735.0357-2023.

Response 4: Thank you for pointing this out. We agree with the comment, and the provided references have been added in the manuscript (Page 6, line 201-203)

Comment 5: A review for active case finding for tuberculosis in migrant populations could be mentioned (8).

8.Pramono JS, Ridwan A, Maria IL, Syam A, Russeng SS, Syamsuar, Mumang AA. Active case finding for tuberculosis in migrants: a systematic review. Med Arch. 2024;78(1):60-64. doi: 10.5455/medarh.2024.78.60-64.

Response 5: Thank you for pointing this out. This has been included in the manuscript (Page 7, line 251-253)

Round 2

Reviewer 1 Report

Comments and Suggestions for Authors

Thank you for responding to the comments. I have one suggestion for the authors to consider: Since the discussions surrounding the barriers to effective TB control and strategies for strengthening TB control largely focus on socially vulnerable groups, the authors might highlight specific categories of migrants who are the most vulnerable to TB and therefore warrant special attention.

Author Response

Reviewer 1

"Thank you for responding to the comments. I have one suggestion for the authors to consider: Since the discussions surrounding the barriers to effective TB control and strategies for strengthening TB control largely focus on socially vulnerable groups, the authors might highlight specific categories of migrants who are the most vulnerable to TB and therefore warrant special attention."

Response 1: Thank you for pointing this out. We agree with the comment and accordingly we have added specific categories of migrants who are the most vulnerable to TB and therefore warrant special attention (Page 5, Line 74-77 – Blue colour)

Reviewer 2 Report

Comments and Suggestions for Authors

The authors have well addressed all the suggestions of our review.

There is a small mistake in the reference 43. The names of the authors appear along with numbers.

Author Response

No comments have been provided